# Depositing a Titanium Coating on the Lithium Neutron Production Target by Magnetron Sputtering Technology

**DOI:** 10.3390/ma14081873

**Published:** 2021-04-09

**Authors:** Zhaopeng Qiao, Xiaobo Li, Yongsheng Lv, Yupeng Xie, Yaocheng Hu, Jie Wang, Haipeng Li, Sheng Wang

**Affiliations:** 1Department of Nuclear Science and Technology, School of Energy and Power Engineering, Xi’an Jiaotong University, No. 28, Xianning West Road, Xi’an 710049, China; qiaozhaopeng@stu.xjtu.edu.cn (Z.Q.); alexlee7@stu.xjtu.edu.cn (X.L.); luyongchee@stu.xjtu.edu.cn (Y.L.); xieyupeng@stu.xjtu.edu.cn (Y.X.); hyc1997@stu.xjtu.edu.cn (Y.H.); wangjie1@xjtu.edu.cn (J.W.); lihaipeng@xjtu.edu.cn (H.L.); 2RIKEN Center for Advanced Photonics, RIKEN, Wako, Saitama 351-0198, Japan

**Keywords:** CANS, lithium target, lithium anticorrosion, titanium coating

## Abstract

Lithium (Li) is one of the commonly used target materials for compact accelerator-based neutron source (CANS) to generate neutrons by ^7^Li(p, n)7 Be reaction. To avoid neutron yield decline caused by lithium target reacting with the air, a titanium (Ti) coating was deposited on the lithium target by magnetron sputtering technology. The color change processes of coated and bare lithium samples in the air were observed and compared to infer the chemical state of lithium qualitatively. The surface topography, thickness, and element distribution of the coating were characterized by SEM, EDS and XPS. The compositions of samples were inferred by their XRD patterns. It was found that a Ti coating with a thickness of about 200 nanometers could effectively isolate lithium from air and stabilize its chemical state in the atmosphere for at least nine hours. The Monte Carlo simulations were performed to estimate the effects of the Ti coating on the incident protons and the neutron yield. It turned out that these effects could be ignored. This research indicates that depositing a thin, titanium coating on the lithium target is feasible and effective to keep it from compounds’ formation when it is exposed to the air in a short period. Such a target can be installed and replaced on an accelerator beam line in the air directly.

## 1. Introduction

Compact accelerator-based neutron sources (CANS) are widely used in neutron image [1,2,3], boron neutron capture therapy [4,5,6], and other fields. Lithium (Li) is one of the commonly used target materials for CANS to produce neutrons by ^7^Li(p,n)7 Be reaction [7,8,9]. However, like other alkali metals, lithium reacts with moisture and oxygen easily. It forms lithium hydroxide (LiOH and LiOH·H_2_O) and other compounds immediately upon exposure to the air [10]. The compounds’ formation will cause a decline of neutron yield. On the other hand, lithium has a low melting point of 180.6 °C. The reaction generates a lot of heat and it may cause lithium evaporation [11,12,13]. To overcome the high chemical activity of lithium, X.-L. Zhou et al. studied the substitution of pure lithium with lithium compounds, such as LiF, Li_2_O, LiH, LiOH, and Li_3_N, and found that the neutron yields decreased by 30% or more [14]. To tackle lithium evaporation, S. Ishiyama et al. synthesized lithium nitride on the surface of the lithium target by in situ nitridation techniques, due to its thermal stability of up to 1086 K [15,16]. But they did not characterize the chemical state of the lithium after the nitridation lithium target was exposed to the air. The anti-evaporation effect also needs to be further tested by proton irradiation experiments. To solve the problems mentioned above, Yoshiaki Kiyanagi et al. attached a thin, metal plate to the target by hot isostatic pressing to make a sealed lithium target [17,18]. The plate with a thickness of a few microns could keep lithium away from the air and confine Li and Be-7 into the target. They have made some lithium sealing tests and no damage was observed on the Ti foil after proton beam irradiation. However, for a given proton energy, the metal plate will cause an evitable energy loss on incident protons, which lead to a decline of the neutron yield.

To protect lithium from the air, we considered depositing a thin, anticorrosion, metal coating on the lithium target. Such a lithium target could avoid compound formation and decrease the neutron yield as little as possible. The coating could also avoid lithium evaporation. The physical vapor deposition (PVD) is proven to be a simple, low-cost, and green technology that has been widely utilized to coat metals and alloys with protective films [19,20]. PVD includes magnetron sputtering, vacuum evaporation, ion plating, etc. Magnetron sputtering was selected in this study for its advantages of high speed, outstanding adhesion, easy control of film thickness, and good film formation property [21,22]. Besides, its low deposition temperature prevents lithium melting during the coating process. As for the coating material, we made comparison among with aluminum [23], chromium [24], and titanium [25,26,27,28]. Finally, titanium (Ti) was selected in this research for its high mechanical strength, good thermal stability, and excellent corrosion resistance.

In this research, a titanium coating of about 200 nanometers thick was deposited on the lithium target by magnetron sputtering technology. The color change processes of coated and bare lithium samples in the air were observed and compared to infer the chemical state of lithium qualitatively. The chemical compositions of the coating were analyzed by the X-ray photoelectron spectroscopy (XPS). The surface topography and thickness of the coating were characterized by scanning electron microscope (SEM). The Ti element distribution on the surface of the coated samples was scanned by energy dispersive spectroscopy (EDS). The lithium compounds were inferred by their X-ray diffraction (XRD) patterns. The Monte Carlo simulations were made to estimate the effects of the Ti coating on incident protons and the neutron yield.

## 2. Materials and Methods

### 2.1. Samples

Figure 1 is a schematic diagram of a lithium sample (1 cm × 1 cm × 190 µmt) used in this work. It consisted of a lithium layer (Tianqi Lithium, Inc. Chongqing, China) with a thickness of 90 µm and a tantalum substrate (Junuo Metal Co., Ltd., Baoji, China) with a thickness of 100 µm, which were combined by the rolling process. There were five of the same lithium samples and one silicon sample shown by this work and numbered from 1–6, respectively. Their components, treatment, and characterization are listed in Table 1. Sample (1) remained a control group without a coating. Samples (2–6) were all subjected to the same coating treatment. The substrate of sample (6) was cut from the p-type (100) Si wafers. It was ultrasonically cleaned in acetone, absolute ethyl alcohol, and deionized water, respectively, for 20 min before coating deposition. Compared with the Li/Ta substrate, silicon substrate is brittle and much easier to get a cross section for the coating thickness measurement. Therefore, we used sample (6) as a reference to assess the average thickness of the Ti coating on samples (2–5).

### 2.2. Magnetron Sputtering

A DC magnetron sputtering system (Chuangshiweina Technology Co., Ltd., Beijing, China) with a titanium target (ϕ75 mm × 5 mm) was used to synthesize the Ti coating on samples (2–6). The purity of the target was over 99.99%. The Ti coating was deposited on the lithium surface of samples (2–5). In order to ensure the accuracy of the control group, sample (1) was transported into the coating chamber in which the sample was left uncoated. The deposition was carried out at room temperature (25 °C) and the maximum temperature of the substrate during the sputtering process was 50 °C. It was much lower than the melting point of lithium (180.6 °C). The DC power, bias, argon flow, working pressure, and base pressure were set as about 100 W (320 mA, 310 V), −70 V, 30 sccm, 0.5 Pa, and 6 × 10^−4^ Pa, respectively. The distance between target to substrate was about 10 cm. The deposition time in this experiment was 55 min. The film thickness was approximately proportional to the sputtering time under the constant pressure and sputtering current, so the thickness could be easily controlled by adjusting the deposition time [26].

### 2.3. Air Exposure Process

According to references [10,16,29], lithium rapidly tarnished, forming a black coating of lithium hydroxide, lithium nitride, and lithium carbonate in the moist air and then became white slowly. So, we could infer qualitatively that there was a reaction between the lithium and air by its color change. After being coated, sample (2) and sample (3) were exposed in the air with a relative humidity of 50% and a temperature of 25 °C. Sample (1) was treated in a same exposure process at the same time. Sample (2) was scratched with tweezers at different positions every 3 h to remove the Ti coating and thus to expose the lithium under the film. The color change of the samples and the scratches were observed and recorded. X-ray diffraction was performed to sample (3) at regular intervals during its exposure to measure whether the titanium-coated lithium had reacted with air and what the derivatives were.

### 2.4. Characterization and Analysis

The chemical composition of the surface of sample (4) was studied immediately when the coating was performed, by an ESCALAB 250 XPS (Thermo, Waltham, MA, USA) with Al Kα X-rays. The binding energy was calibrated by C 1s peak of 284.8 eV. The surface morphology of samples (1) and (5) and the cross-sectional morphology of the silicon sample (6) were captured by a 7800F Schottky field SEM (JEOL, Tokyo, Japan). The titanium distribution on the surface of sample (5) was observed by an Oxford EDS (Oxfordshire, UK). To verify the composition change of samples (1) and (3) after exposure in the air, X-ray powder diffraction was performed on them by a PANalytical X’Pert Pro diffractometer (Almelo, The Netherlands) with Cu Kα radiation. The composition information, though not shown in XRD, could be inferred based on the XRD patterns, since the derivatives of lithium and air were known [10,16,29]. The reason why we used XRD instead of XPS is that the latter could only be performed in a high-vacuum environment. However, we found that lithium samples that had reacted with air outgassed a lot under the low pressure, which made it difficult and time consuming to create a vacuum condition. By contrast, XRD test was performable in the atmosphere.

### 2.5. Monte Carlo Simulation

Theoretically, the titanium film in front of the lithium will cause energy loss of protons, which leads to a decrease of the neutron yield. In order to explore these impacts, taking 2.5 MeV incident protons as an example, the Monte Carlo simulation was made by the stopping and range of ions in matter (SRIM) [30] code and Monte Carlo N-Particle (MCNP) [31] code version 6. The calculation models consisted of a beam of protons of 2.5 MeV, a two-layer target, a titanium layer, and a lithium layer. The proton beam was perpendicular to the target surface. The lithium density was set as 0.534 g/cm^3^. The titanium thickness ranged from 0–1000 nm. The thickness of lithium was set as constant at 90 µm, which could reduce the mean energy of a proton beam from 2.5 MeV to about 1.88 MeV, the ^7^Li(p,n)7 Be reaction threshold [32]. The relationship among proton energy loss, neutron yield, and titanium thickness was investigated.

## 3. Results and Discussion

### 3.1. Samples with/without Coating in the Air

Figure 2 shows the photograph of sample (1) and sample (2). Sample (1) in the air changed from metallic silver to homogeneous dark gray in a few minutes, and was dappled with black and white one hour later. After three hours, it gradually turned off-white, then slowly turned pure white, and stopped changing afterwards. This process of color change was due to the reaction of lithium with air [10]. Samples (2) and (3) changed from metallic silver to uniform orange, brown, and gray, successively, within a few minutes, and then stopped changing afterwards. Such a change was caused probably by the reaction of titanium coating with air. Then, 3 h later, sample (2) was scratched by tweezers and it was found that the coating had outstanding adhesion with the lithium substrate. The scratch was silver at first, then quickly became dark gray, and gradually turned off-white. This process of change was very similar to that of the lithium sample (1) in the air. It can be inferred that the coated lithium of sample (2) did not deteriorate after being exposed to the air for 3 h. The same scratch test was performed to sample (2) every 3 h, and we found that with 9-h exposure, the color change of the scratch was still consistent with the above situation.

### 3.2. XPS Results

After being coated, sample (4) was immediately transferred into the chamber of XPS in situ. The wide scan was performed to survey the chemical elements of the sample surface to check out whether the lithium was evenly covered with Ti coating. The scanning area was located in the middle of the sample. The XPS spectra are shown in Figure 3 and the 20–80 eV region is shown as a small inset of it. Ti peaks were observed as expected. The Li 1s peak was located around 54 eV [13,33], and no obvious peak was found near this region, which means that a titanium coating was successfully deposited on the lithium surface and the lithium was coated completely. In view of the inelastic scattering mean free path (IMFP) of the photoelectron [34,35], the thickness of Ti coating was fairly thicker than a few nanometers. In addition, the oxygen peak also appeared. It was because the titanium target was inevitably oxidized slightly when it was assembled in the deposition chamber. Besides, it was also probably caused by the trace air in the chamber.

### 3.3. SEM and EDS Results

The surface morphologies of samples (1), (5), and (6) are shown as Figure 4a–c, respectively. Sample (1)’s surface was rough and pitted with fissures of varying sizes. Sample (5)’s surface was undulating, but smoother than that of (1) and fewer fissures were found. The surface of sample (6) was very flat and with no fissure. The flatness difference between sample (5) and sample (6) was caused by their different substrates. The surface of the silicon substrate was mirror-like and flatter than that of the lithium substrates. A grainy structure was observable on the surfaces of sample (5) and sample (6). But the grain size of (5) was larger than that of (6). This difference was also caused by the different surface roughness of their substrates that effected the nucleation and microstructural growth of Ti coating [36,37]. Nonetheless, the surface morphology of sample (5) was obviously different from sample (1) but similar to sample (6). The EDS mapping was performed on sample (5) to search Ti element distribution. The scan place was the same as where Figure 4b shows. It was found that Ti element had a uniform distribution on the surface of sample (5), as shown in Figure 5a. To estimate the Ti coating thickness, the SEM cross section of sample (6) was photographed, as shown in Figure 5b. The thickness was about 200 nanometers. From the XPS, SEM, and EDS results, it was considered that the titanium coating had good property in coating formation on the lithium sample surface. The surface of samples (2–6) were successfully deposited with a uniform titanium film of about 200 nm thick.

### 3.4. XRD Results

An X-ray diffraction analysis was performed on sample (1) after it was exposed to the air for 3 h. The patterns are shown in Figure 6, with obvious diffraction peaks of lithium hydroxide and weak diffraction peaks of tantalum, indicating that sample (1), after three hours of exposure in the air, reacted with air and became LiOH, and thus the color turned off-white from metallic silver. The scratch test in Section 3.1 showed that the lithium of sample (2), after nine hours of exposure, had no reaction with air. To verify that, XRD was performed to sample (3) at this time. The pattern is shown in Figure 6, from which it can be seen that, except for strong diffraction peaks of lithium, no peak of LiOH appeared. That means lithium was well protected in the first 9-hour exposure in the air. Some weak peaks were also found and they were corresponding to titanium or tantalum. To tell which one it was, XRD was also performed on sample (6) and no Ti peak was found. So, the weak peaks were Ta peaks, and the Ti coating we prepared was amorphous. After being exposed to the air for 17, 26, and 46 h, sample (3) was further characterized by XRD three times, with the results shown in Figure 6. From the patterns, we could tell that with the extension of the exposure time, the lithium peaks weakened and the LiOH peaks appeared and increased gradually. After a 46-hour exposure, the XRD pattern of sample (3) was quite similar with that of sample (1). Both of them showed strong peaks of lithium hydroxide and no lithium peak was found. That means lithium was completely decomposed. These results indicated that sample (3) stayed stable in the first nine hours and gradually deteriorated afterwards when it was exposed to the air.

### 3.5. Monte Carlo Simulation Results

Taking 2.5 MeV incident proton as an example, the relationship between the thickness of the titanium coating and the energy loss of the proton was calculated by SRIM code, as shown in Figure 7a. Within a certain range, as the thickness of the titanium coating increased, the resulting proton energy loss also increased linearly. For a 200-nm Ti coating, the corresponding energy loss of 2.5 MeV incident proton was about 7 keV, accounting for about 3‰ of the total. So theoretically, energy loss of a thin titanium film of hundreds of nanometers on protons is very small. The thinner the coating, the smaller the effect on the protons. However, a coating might be too thin to isolate a lithium target from the air. In addition, on the premise that the lithium could be effectively protected, an over-thick coating was unnecessary, for it would cause a larger energy loss to protons. The neutron yield of the lithium target with a single-side, coated, Ti film was calculated by MCNP6, with the result shown in Figure 6. As the thickness of the titanium film increased, the neutron yield gradually decreased. The neutron yield of a lithium target with a 200-nm Ti coating was 99.86% of that of pure lithium. So, the effect of a thin Ti coating on the neutron yield was extremely limited. To sum up, when protons with 2.5 MeV energy bombard a 200-nm Ti-coated lithium target, the effects of Ti coating on the protons and the neutron yield are little and negligible.

## 4. Conclusions

In this paper, a research of thin anticorrosion coating on lithium target for CANS was carried out. It is the first time depositing a coating on the lithium target by magnetron sputtering technology was carried out. A bare lithium sample and five titanium-coated ones were studied in this research. The corrosion of bare lithium in the air was determined qualitatively according to color change. The surface chemical elements, morphology, and compositions of the samples were characterized by XPS, EDS, SEM, and XRD measurements to evaluate the protection effect of Ti coating on Li target. The influences on the incident proton beam and the neutron yield caused by the Ti coating were estimated by the Monte Carlo simulation. Based on the above work, the results were derived as follows:

(1) The corrosion of bare lithium in the air happened quickly, and the corrosion product after 3 h of exposure was mainly LiOH.

(2) By magnetron sputtering technology, thin Ti anticorrosion coating can be plated on the Li target surface.

(3) A 200-nm Ti coating can effectively isolate Li from the air and stabilize its chemical state for at least 9 h, at a relative humidity of 50% and a temperature of 25 °C.

(4) Taking 2.5 MeV incident protons as an example, the simulation showed that energy loss rate of 200-nm Ti film for protons was about 3‰, and the reduction rate of the neutron yield was less than 2‰.

From these results, it can be concluded that depositing a thin titanium coating on the lithium target by magnetron sputtering technology is feasible. Such a thin Ti coating is effective to prevent lithium from deteriorating reaction with air in a short period. The influence of a 200-nm, thin, Ti coating on the incident protons and the neutron yield can be ignored. The lithium target coated with titanium is more convenient to store and transport than a bare lithium target, and it can be directly installed and replaced on an accelerator beam line in the air, instead of carrying out in a vacuum or an ultra-low humidity environment. Furthermore, the titanium coating may also be able to avoid lithium evaporation and seal the radionuclide Be-7 produced by ^7^Li(p,n)7 Be, which needs to be verified in further experiments.

## Figures and Tables

**Figure 1 materials-14-01873-f001:**
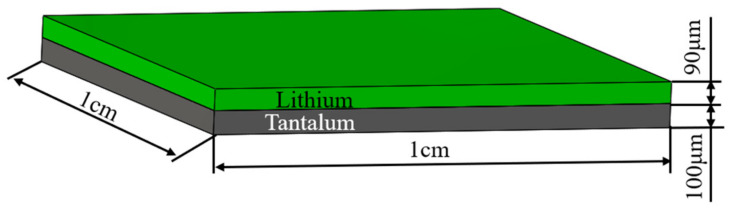
A schematic view of the lithium sample. The lithium surface to be coated with titanium.

**Figure 2 materials-14-01873-f002:**
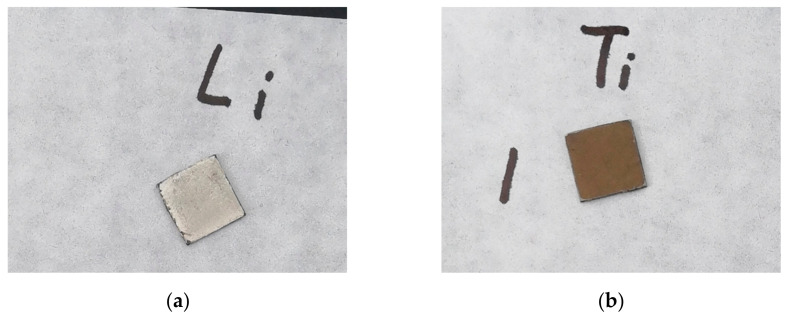
Photograph of sample (1) and sample (2): (**a**) sample (1) after being exposed to the air for a minute; (**b**) sample (2) after being exposed to the air for a minute.

**Figure 3 materials-14-01873-f003:**
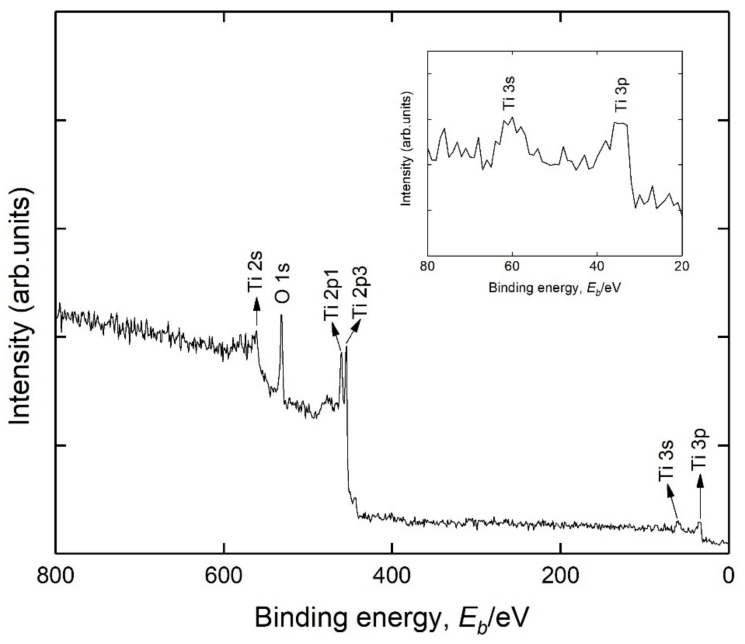
Wide-scan XPS spectra of sample (4).

**Figure 4 materials-14-01873-f004:**
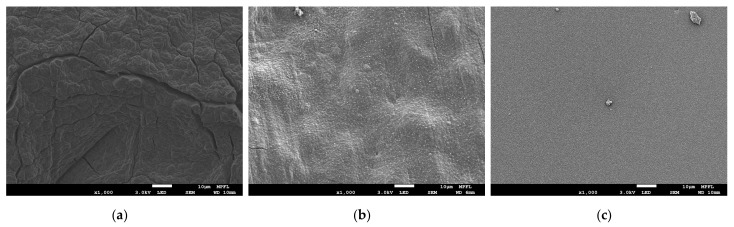
The SEM image of samples (1), (5), and (6). (**a**) Sample (1), bare lithium sample. (**b**) Sample (5), Ti-coated lithium sample. (**c**) Sample (6), Ti-coated silicon sample.

**Figure 5 materials-14-01873-f005:**
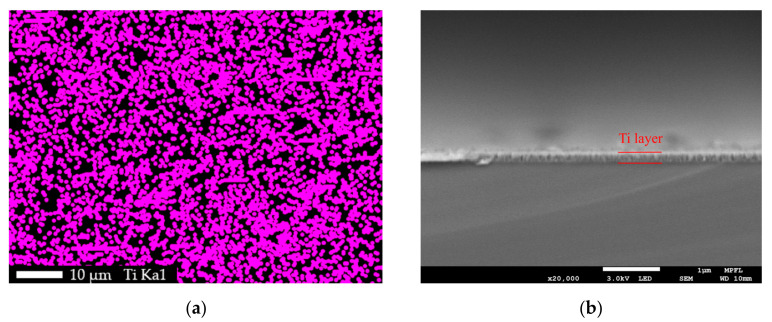
Titanium distribution and its thickness. (**a**) EDS mapping of Ti element distribution on the surface of sample (5). (**b**) The SEM cross-section image of sample (6) with a coating thickness of about 200 nm.

**Figure 6 materials-14-01873-f006:**
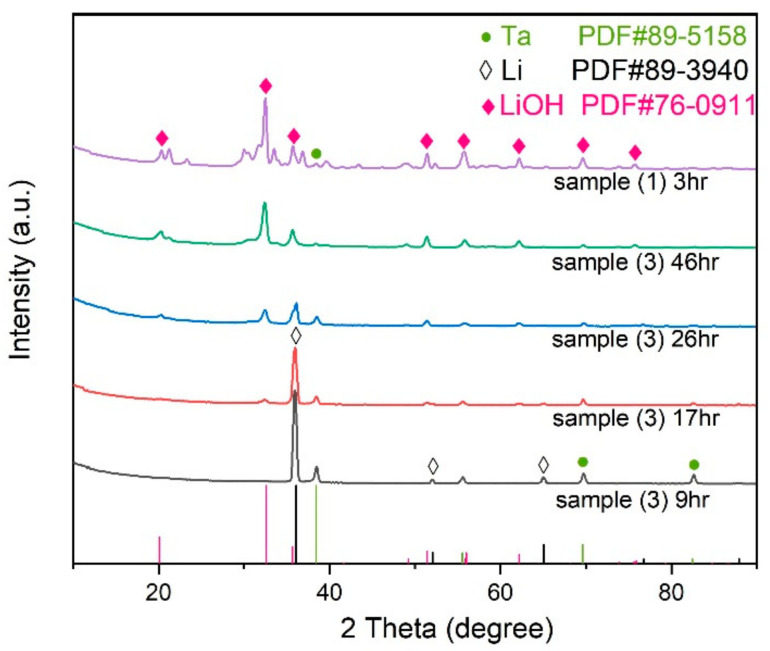
The XRD patterns of sample (1) and sample (3) after different exposure times in the air. After three hours, the LiOH peaks of sample (1) were obvious and no lithium peaks were shown. After nine hours, only the lithium peaks and tantalum peaks could be found in the diffraction pattern of sample (3). After that, LiOH peaks began to appear and increased gradually with the extension of the exposure time.

**Figure 7 materials-14-01873-f007:**
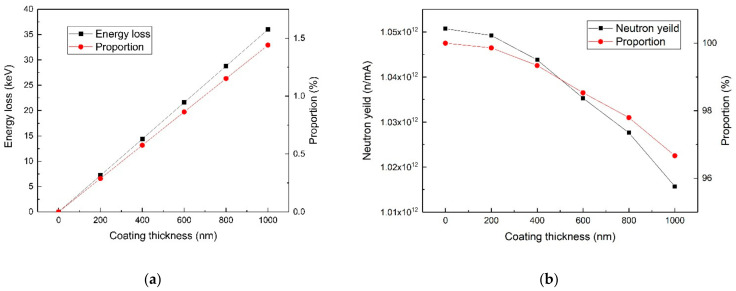
The Monte Carlo simulation results. (**a**) The relationship between the proton energy loss and titanium thickness. (**b**) The relationship between the neutron yield and titanium thickness.

**Table 1 materials-14-01873-t001:** Sample numbers, components, treatment, and characterization methods.

Number	Substrate Components	Coating	Treatment	Characterization
(1)	Li/Ta	-	Exposure	Observation; SEM; XRD
(2)	Li/Ta	Ti	Exposure; Scratch	Observation
(3)	Li/Ta	Ti	Exposure	XRD
(4)	Li/Ta	Ti	-	XPS
(5)	Li/Ta	Ti	Exposure	SEM; EDS
(6)	Si	Ti	Section	SEM; XRD

## Data Availability

Data available on request from the authors.

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
