# Peer review of "Depositing a Titanium Coating on the Lithium Neutron Production Target by Magnetron Sputtering Technology"

_materials, 2021, doi:10.3390/ma14081873_

Round 1

Reviewer 1 Report

The paper deals with one approach of technological basis for development of Li target for CANS. To reviewer's opinion is that Introduction is too short.  It is not clear especially for non-specialist which type of Li target the data presented are relevant. Authors should give more details when they referring to paper under [ref 16]. In adition reviewer suggest editing the paper title. Please consider as example "Air corrosion of Li covered with sputter deposited Ti films".

line 82   2.2. Magnetron sputtering
As protection with Ti coating is a central point of the paper, the sputter method should be given in more details: discharge voltage, discharge current, substrate to target distance, bias applied to substrate are all useful for reproducing the result. 

line 164     Fig.3
It should be shown the spectrum region 0-100 eV (similar to ref. 14) and arrow where Li 1s. 

line 187    Fig.5a
Reviewer could not understand what is shown in this picture and how he might lern that Ti is distributed uniformly. It is also unclear why there are no data on EDS measurents of Ti amount (thickness). Being no SEM operator, Reviewer often ask and get an answer that LayerProbe paket (oxfod instruments software) is not activated. It is only advice that Reviewer uses free software - GMRFilm  as axample. As the substrate is low Z element other, more recenly free software could also be used. 

Author Response

Dear reviewer:

Thanks a lot for your careful review of the manuscript. All the changes in the manuscript are marked in red and here is my response.

Point 1: To reviewer's opinion is that Introduction is too short. It is not clear especially for non-specialist which type of Li target the data presented are relevant. Authors should give more details when they referring to paper under [ref 16]. In adition reviewer suggest editing the paper title. Please consider as example "Air corrosion of Li covered with sputter deposited Ti films".

Response 1:

  1. The content about lithium change in the air is added in the introduction. The problems of lithium target are explained more clearly. They are shown Line 32-36.
  2. The work of other scientists is written more specifically now, especially about the sealed lithium target of reference [16], which is shown Line 43-50. The new serial number of the reference [16] is [17].
  3. The original title “research on depositing a titanium coating …” is indeed not accurate enough. It sounds like the coating itself is the key point. In fact, the critical thing and purpose of this work is to improve anticorrosion of lithium target. On the other hand, the coating may also be able to avoid lithium evaporation and seal the radionuclide Be-7. Besides, we didn’t focus on the mechanism of Li air corrosion and Ti film in this work. So for the time being, we changed the title as “depositing a titanium coating on the lithium neutron production target by magnetron sputtering technology”.

Point 2: line 82   2.2. Magnetron sputtering

As protection with Ti coating is a central point of the paper, the sputter method should be given in more details: discharge voltage, discharge current, substrate to target distance, bias applied to substrate are all useful for reproducing the result.

Response 2: Yes. The parameters of depositing Ti film in this work are all listed in the paper now. The discharge voltage, discharge current, substrate to target distance and bias was 310 V, 320 Ma, 10 cm and -70 V respectively. It is shown Line 97-100.

Point 3: line 164     Fig.3

It should be shown the spectrum region 0-100 eV (similar to ref. 14) and arrow where Li 1s.

Response 3: Yes. The Li 1s peak is located around 54 eV. To see more clearly, we change the region of 0-100 eV to 20-80 eV. The 20-80 eV region is shown as small inset of Figure 3. No obvious peak was found near 52-56 eV. So we think the lithium was coated completely. It is also indicated on line 165-167.

Point 4: line 187    Fig.5a

Reviewer could not understand what is shown in this picture and how he might lern that Ti is distributed uniformly. It is also unclear why there are no data on EDS measurents of Ti amount (thickness). Being no SEM operator, Reviewer often ask and get an answer that LayerProbe paket (oxfod instruments software) is not activated. It is only advice that Reviewer uses free software - GMRFilm  as axample. As the substrate is low Z element other, more recenly free software could also be used.

Response 4:

  1. EDS has several modes. We performed the element mapping mode to the sample (5). It can be used to approximate the enrichment region of certain elements. Under this mode, the distribution of Ti elements would be found and marked as purple dots. From the Figure 5(a), it can be seen purple dots (Ti element) have a uniform distribution. There was some blank area could also be found. Because we just repeated the scan for 4 times and more Ti will be found if we repeated the scan more times. In fact, considering all the XPS, SEM and EDS results, it is more convincing that the titanium has a uniform distribution on the sample (2-5) surface.
  2. In this work, EDS was only for qualitative judgment of Ti element distribution. And EDS could only give a relative account of element, not an absolute value. The coating thickness was measured by its cross-section with SEM. EDS is not applicable to measure thickness. The reason was shown Line 84-87. The thickness is shown in Figure 5(b).
  3. We use Oxford instrument indeed. And the software is INGA, it is not so convenient for personal users. Thanks for telling me the free GMRFilm. I will try it on my PC.
    Best regards!   Zhaopeng Qiao

Reviewer 2 Report

The authors have resubmitted the previous manuscript materials-1151761, presenting a study on decline caused by lithium target reacting with the air, a titanium (Ti) coating was deposited on the lithium target by magnetron sputtering technology. lithium sample and 5 titanium coated ones were studied in 239 this research. Although the authors have include some aditional information (some pictures if the experimental setup and some explanatory paragraphs), I don’t see any improvement or substantial change in the manuscript.
As I pointed out in my previous review, the use of English must be carefully revised. As it is written, the manuscript is really difficult to follow and some sentences are meaningless. Not only the use of English in the original text has not been revised, but the new included paragraphs contain similar mistakes. 
In view of this new revised version, I have to confirm my previous report and recommend major revision of the manuscript. 

Author Response

Dear reviewer:

Thanks a lot for your review. All the changes in the manuscript are marked in red and here is my respons.

Point1: The authors have resubmitted the previous manuscript materials-1151761, presenting a study on decline caused by lithium target reacting with the air, a titanium (Ti) coating was deposited on the lithium target by magnetron sputtering technology. lithium sample and 5 titanium coated ones were studied in 239 this research. Although the authors have include some aditional information (some pictures if the experimental setup and some explanatory paragraphs), I don’t see any improvement or substantial change in the manuscript

Response 1: Sorry, a little confused. In fact, this is the first time I received your review report, and it is the second time I submit the manuscript materials-1151761.

Point 2: As I pointed out in my previous review, the use of English must be carefully revised. As it is written, the manuscript is really difficult to follow and some sentences are meaningless. Not only the use of English in the original text has not been revised, but the new included paragraphs contain similar mistakes. 
In view of this new revised version, I have to confirm my previous report and recommend major revision of the manuscript. 

Response 2: In this 2nd edition, we did the following modifications:

  1. The content about lithium change in the air is added in the introduction. The problems of lithium target are explained more clearly in the introduction. It is shown Line 32-36.
  2. The work of other scientists is written more specifically now, which is shown in the first paragraph of introduction.
  3. The parameters of depositing Ti film in this work are all listed in the paper now. It is shown Line 97-100.
  4. The 20-80 eV region is shown as small inset of Figure 3, to see more clear whether there is peak of Li 1s.
  5. Some sentences are rewritten, such as line 119 and line 123-125.          
  6. Some other minor changes that are marked in red.
  We check the content of the article again, especially the grammatical expression. Please check this 2nd edition     Best regards!   Zhaopeng Qiao

Reviewer 3 Report

General consideration

Zhaopeng Qiao and coauthors describe the effect of the application of a titanium coating on a lithium substrate used as target material for compact accelerator-based neutron source (CANS). In the introduction they explore the limitations of this material due to the rapid decomposition of lithium target and they propose as a solution a titanium coating deposited with the magnetron sputtering technology.

The article is well presented and written, the topic is interesting and innovative and the obtained results are interesting and promising for further application.

My suggestion for the editor is to publish this work after some small revisions.

Introduction

Line 32: you explain the rapid and easy degradation of lithium target with “(lithium) easy reaction with air to form compounds”. In this case I'd rater that you explain better the concept: lithium doesn’t react with air, eventually it reacts with moisture, oxygen and so on. Over that, you examined in the results the compounds that are formed after degradation such as hydroxide, you can introduce here the formation of these compounds. In other words you have to examine in depth the degradation of lithium, that is the main reason that pushes you to work on this topic.

Materials and methods

Line 111: the phrase “the binding energy was calibrated by C1s peak of 284.8 eV.” is referred to the XPS analysis. You have to move it after the description of the instrument. At the moment is after the description of the SEM. 

Line 114-116: I found the sentence concerning the XRD really confusing, consider to rewrite it.

Results and discussion

Line 207: I think that the word “destroyed” is not appropriate, probably “decomposed” is better.

Author Response

Dear reviewer:

Thanks a lot for your careful review of the manuscript. All the changes in the manuscript are marked in red and here is my response.

Point 1: Line 32: you explain the rapid and easy degradation of lithium target with “(lithium) easy reaction with air to form compounds”. In this case I'd rater that you explain better the concept: lithium doesn’t react with air, eventually it reacts with moisture, oxygen and so on. Over that, you examined in the results the compounds that are formed after degradation such as hydroxide, you can introduce here the formation of these compounds. In other words you have to examine in depth the degradation of lithium, that is the main reason that pushes you to work on this topic.

Response 1: The content about lithium change in the air is added in the introduction. And the problems of lithium target are explained more clearly. They are shown Line 32-36. Maybe the lithium corrosion is still not detailed enough in this manuscript for the time being. Because we think the key point and purpose of this work is to improve anticorrosion of lithium target. For this work, whether there is a change of lithium state is the most important thing, and what it becomes is not so important. If necessary, the mechanism of Li corrosion in air will be explained more carefully in the introduction.

Point 2: Line 111: the phrase “the binding energy was calibrated by C1s peak of 284.8 eV.” is referred to the XPS analysis. You have to move it after the description of the instrument. At the moment is after the description of the SEM.

Response 2: Yes. It should be move forward. It is shown Line 119 now.

Point 3: Line 114-116: I found the sentence concerning the XRD really confusing, consider to rewrite it.

Response 3: The sentence was rewritten as ‘X-Ray diffractometer (XRD) with Cu Kα radiation was performed to them on a X'Pert Powder X-Ray diffractometer’. It is shown Line 123-125.

Point 4: Line 207: I think that the word “destroyed” is not appropriate, probably “decomposed” is better.

Response 4: The “destroyed” has been changed into “decomposed”. On line 219 now.

That's all.

Best regards!

Zhaopeng Qiao

Round 2

Reviewer 1 Report

Reviewer accept all additions made by authors to the text. Concerning Fig.5a he want say the following. In evaluation of Ti film as a protection coating a key point is the complete (entire) and homogenious covering the Li surface by Ti. Again to Reviewer's opinion the line scan of Ti Ka intensity (instead of the elemental mapping) would improve reading the results as it would tell that (1) there are no Ti free spots and (2) the film thickness is indeed constant as it is seen in Fig. 5b. In addition EDX spectrum of sample 5 (in view of possible impurities) would also improve the presentation.  

Reviewer 2 Report

Dear authors

I have read the present manuscript with interest. I have, however, found that there quite a few points that merit modification before this work can be considered for publication.